# People with COPD have greater participation restrictions than age-matched older adults without respiratory conditions assessed during the COVID-19 pandemic

Sachi O'Hoski[1,2], Ayse Kuspinar[1◉], Joshua Wald[1,3◉], Julie Richardson[1◉], Roger Goldstein[2◉], Marla K. Beauchamp[1,2,3]*

1 School of Rehabilitation Science, McMaster University, Hamilton, ON, Canada, 2 Respiratory Research, West Park Healthcare Centre, Toronto, ON, Canada, 3 Firestone Institute for Respiratory Health, St. Joseph's Healthcare, Hamilton, ON, Canada

◉ These authors contributed equally to this work.
* beaucm1@mcmaster.ca

## Abstract

### Background

Participation restriction has detrimental effects for older adults but it is unknown how participation differs for people with chronic obstructive pulmonary disease (COPD) compared to older adults of the same age without respiratory conditions. We compared scores on the Late Life Disability Instrument (LLDI) between people with COPD (study group) and a random sample of older adults (control group).

### Methods

Participants with COPD (study group) were recruited from two hospitals in Ontario and age- and sex-matched with a ratio of 1:2 with participants from a random sample of community-dwelling older adults who did not report having respiratory conditions (control group). The study group completed the LLDI prior to the COVID-19 pandemic and the control group completed the LLDI at the end of the first wave of the pandemic. LLDI frequency and limitation scores were compared between groups using Wilcoxon rank-sum tests.

### Results

Forty-six study group participants (mean age 74.2 (SD 5.5) years) and 92 control group participants (mean age 74.4 (SD 5.4) years) were included. Fifty-four percent of the participants were female. The majority of the study group had severe COPD (median forced expiratory volume in one second of 34.5 (25th-75th percentile 27.0–56.0) % predicted). LLDI sores were lower for the study group compared to the control group for both the frequency (median difference -5.4 points, p<0.001) and limitation (median difference -7.6 points, p<0.001) domains. The personal subscale demonstrated the largest magnitude of difference between

**Data Availability Statement:** All relevant data are within the paper and its Supplementary files.

**Funding:** SO is funded by a Vanier Canada Graduate Scholarship and MKB is supported by a Canada Research Chair in Mobility, Aging and Chronic Disease (Tier 2). The primary studies were completed with the support of an Ontario Respiratory Care Society Grant and a Labarge COVID-19 Grant from the McMaster Institute for Research on Aging. At the time of data collection SO was also supported by an Ontario Respiratory Care Society fellowship and a Breathing as One Fellowship from the Canadian Lung Association. The funders had no role in the study design, data collection and analysis, decision to publish, or preparation of the manuscript.

**Competing interests:** The authors have declared that no competing interests exist.

groups (median difference -13.4 points) and the social subscale demonstrated the smallest magnitude of difference (-5.2 points).

## Conclusion

People with COPD had greater participation restrictions than a random sample of older adults without ongoing respiratory conditions. The differences seen in participation between the two groups may have been reduced due to temporal confounding from the COVID-19 pandemic. While participation is relevant to all older adults, our results suggest that it is especially important that it be assessed in those with COPD.

## Introduction

Chronic obstructive pulmonary disease (COPD) is a highly prevalent condition worldwide [1,2] that is characterized by airborne particulate exposure causing irreversible damage to the lungs. The most common cause is tobacco smoke often in combination with other environmental exposures and/or genetic susceptibility [2,3]. The primary signs and symptoms associated with COPD are breathlessness, cough and increased secretions or sputum production [2]. In addition, people with COPD often present with extra-pulmonary manifestations of their condition, including skeletal muscle dysfunction, reduced exercise tolerance, physical inactivity, functional impairment, reduced quality of life, and social isolation [4]. However, there is little information regarding the impact of COPD on participation.

Participation, as it is conceptualized in the World Health Organization's International Classification of Functioning, Disability and Health [5], is involvement in a life situation, congruent with the concept of disability in both Nagi's disablement model [6] and Verbrugge and Jette's disablement process model [7]. In these models or frameworks, 'participation restriction' or 'disability' arises from the interaction between functional limitations caused by health conditions such as COPD and intrinsic or personal factors such as age and sex, and the environment. It reflects limitations in the ability to perform tasks that are expected given one's role in a specific sociocultural context and physical environment [6].

The prevalence of participation restriction for community-dwelling adults ≥ 50 years has been estimated to be 52% with the most affected area on the Keele Assessment of Participation being mobility outside the home [8]. In a report assessing basic and instrumental activities of daily living and social participation in 6903 adults ≥ 65 years with chronic conditions such as arthritis, ischaemic heart disease and diabetes, up to 68% of the cases of disability would not have occurred if it were not for the presence of these chronic conditions [9]. Given the mediating effect of social participation restrictions on psychological distress [10] and the protective effect of social relationships against mortality [11], there is a clear need to better understand this construct in older adults with chronic conditions.

COPD may present important challenges to participation that are not seen in other chronic conditions because of its progressive symptoms such as breathlessness and cough, the nature of its episodic exacerbations, and devices employed in its management such as mobility aids and supplemental oxygen [2]. However, to our knowledge, there are no studies that have compared scores on a validated measure of participation between people with and without chronic lung disease, limiting our knowledge of the impact of COPD on participation. Such information will assist healthcare professionals in formulating a care plan that addresses this important aspect of health. Therefore, the objective of this study was to compare participation scores in

people with COPD to scores from a random sample of older adults using a widely-used, validated measure of participation, the Late Life Disability Instrument (LLDI) [12]. The LLDI is based on Nagi's disablement model [6], and is consistent with the concept of participation in the international classification of functioning, disability and health [5]. We hypothesized that the study group (those with COPD) would have lower scores than the control group on both domains, meaning greater restriction in the frequency of participation as well as greater perceived limitations in their ability to participate.

## Materials and methods

This was a secondary analysis of data from two studies, the first a cross-sectional study in patients with COPD (conducted from February 2018 to March 2020), and the second, baseline data from a longitudinal study of older adults conducted during the coronavirus disease 2019 (COVID-19) pandemic (May to August 2020). Ethics approval for the primary studies was obtained from the Joint West Park Healthcare Centre—Toronto Central Community Care Access Centre—Toronto Grace Health Centre Research Ethics Board (17-013WP) and the Hamilton Integrated Research Ethics Board (HiREB #3878 and #10814). All participants provided informed consent prior to data collection for the primary studies; written consent was obtained from the study group and verbal consent was obtained from the control group.

### Study group participants

We recruited study group participants during routine clinical visits at two respiratory centres in Ontario- West Park Healthcare Centre in Toronto and the Firestone Institute for Respiratory Health in Hamilton. They were recruited for a cross-sectional validation study of the primary outcome measure, the LLDI. Details of recruitment, eligibility, and data collection have been reported previously [13]. Briefly, participants living in the community had to have a physician diagnosis of COPD as well as a 10-pack-year smoking history.

### Control group participants

A random sample of older adults was identified using 2016 census data and a sampling company that provides representative samples of publically available phone numbers [14]. These participants had been recruited for a longitudinal tele-survey looking at the impact of the COVID-19 pandemic on the mobility and participation of community-dwelling older adults who were not suffering from COVID-19 [15]. Postal codes were selected based on the distance from McMaster University in Hamilton and the ratio of older adults ($\geq$ 65 years) within the dissemination area. Participants had to be living independently within the Greater Hamilton Area, aged 65 or older, and able to provide consent. Potential participants were excluded if they had severe and uncorrectable cognitive, visual or hearing impairments that would prevent their completing the questionnaires. The baseline data was used for this analysis.

### Outcome measure

The disability component of the Late Life Function and Disability Instrument (LLDI) was administered via one-on-one interview to both groups. Respondents were asked how often they participated in 16 various life tasks (frequency scale) and how limited they were in participating in those same tasks (limitation scale) [12]. The frequency scale is comprised of a social role subscale (9 items related to going out with others) and a personal role subscale (7 items related to personal care and local errands). The limitation scale is comprised of an instrumental role subscale (12 items related to moving around the home and community) and a

management role subscale (4 items related to communication and planning) [12]. We used the scaled summary scores with possible scores from 0–100% and higher scores indicating greater frequency of and fewer limitations in participation [12]. The minimal detectable change score ($MDC_{90}$) for the LLDI is 7.4 points for the frequency scale and 11.6 points for the limitation scale in mobility-limited older adults [16] and the $MDC_{95}$ is 6.7 points for the frequency scale and 9.9 points for the limitation scale in people with COPD [13]. This measure has been used extensively in older adults [17] and has shown good construct validity and test-retest reliability in people with COPD [13].

## Data analysis

We performed statistical analyses using Stata 14.2 (StataCorp LLC, College Station, Texas). We explored raw data for normality visually using histograms and numerically using the Shapiro–Wilk test. Based on the distribution of the data, we used either independent students t-tests or Wilcoxon rank-sum tests to compare the study group and the control group. If the F-test for equal variances was significant, unequal t-tests were performed and Satterthwaite's approximation of degrees of freedom reported. We used one-sided tests and applied the Bonferroni correction for multiple comparisons. Raw data is available in S1 File. As a sensitivity analysis, we also compared LLDI scores between the full samples of people with and without respiratory disease, without age- and sex-matching using quantile regression with age and sex as covariates. We accepted an alpha value of $\leq 0.05$ as indicating statistical significance.

## Results

From the primary studies, LLDI scores were collected in 96 people with COPD and 272 older adults. For this analysis, we excluded participants from the control group if they had a physician-diagnosed respiratory condition such as COPD or asthma. Participants were then age- and sex-matched for the study and control groups at a ratio of 1:2, matching age within 2 years. This resulted in 46 study group participants and 92 control group participants (Fig 1).

Participants had a mean age of 74 years and 54% of them were female. The study group participants had a median forced expiratory volume in 1 second of 34.5% predicted,

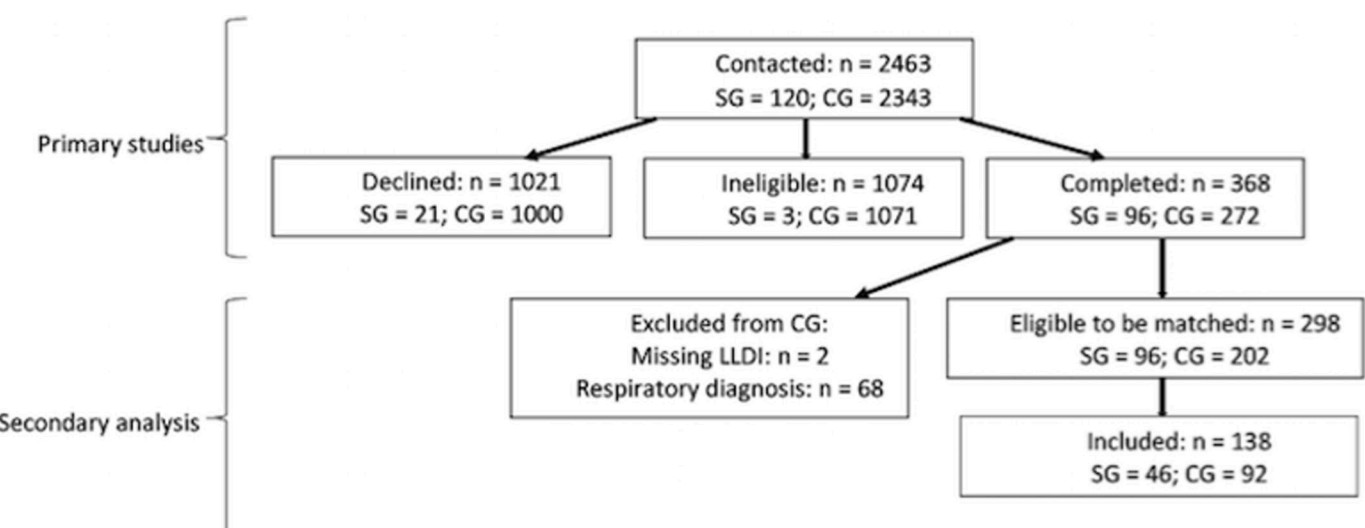

**Fig 1. Flow diagram of recruitment for primary studies and inclusion in current analysis.** SG = study group; CG = control group.

**Table 1. Participant characteristics.**

| | Study Group, n (%)[a], n = 46 | Control Group, n (%)[a], n = 92 | p-value of between-group difference |
|---|---|---|---|
| Age, y, mean (SD) | 74.2 (5.5) | 74.4 (5.4) | 0.84 |
| Sex, female | 25 (54.4) | 50 (54.4) | 1.00 |
| BMI, kg/m$^2$, median (25$^{th}$-75$^{th}$ percentile)[b] | 27.9 (24.1–33.3) | 26.7 (24.4–29.6) | 0.40 |
| Self-reported general health: | | | <0.001 |
| Excellent | 0 (0.0) | 17 (18.5) | |
| Very good | 1 (2.2) | 35 (38.0) | |
| Good | 16 (34.8) | 32 (34.8) | |
| Fair | 18 (39.1) | 7 (7.6) | |
| Poor | 11 (23.9) | 1 (1.1) | |
| Uses Gait Aid | 26 (56.5) | 10 (10.9) | <0.001 |
| Uses Supplemental Oxygen | 29 (63.0) | Not collected | N/A |
| Modified medical research council dyspnea scale, mean (SD)[c] | 2.2 (0.9) | N/A | N/A |
| COPD assessment test, mean (SD)[d] | 21.6 (6.4) | N/A | N/A |
| Most common comorbidities (reported in >20% of total sample): | | | |
| Hypertension | 28 (60.9) | 41 (44.6) | 0.07 |
| Anxiety | 13 (28.3) | 4 (4.4) | <0.001 |
| Back pain | 10 (21.7) | 18 (19.6) | 0.77 |
| Cancer | 10 (21.7) | 16 (17.4) | 0.54 |
| Cataracts | 8 (17.4) | 31 (33.7) | 0.046 |
| Diabetes | 8 (17.4) | 20 (21.7) | 0.55 |
| Osteoarthritis | 8 (17.4) | 17 (18.5) | 0.88 |
| Depression | 8 (17.4) | 6 (6.5) | 0.047 |
| Osteoporosis | 7 (15.2) | 10 (10.9) | 0.47 |

BMI = body mass index; N/A = not applicable; COPD = chronic obstructive pulmonary disease.

[a]Unless stated otherwise.

[b]n = 86 for control group (6 participants did not know their weight).

[c]0-4 points, higher = worse dyspnea.

[d]0-40 points, higher = greater impact of COPD.

corresponding to a global initiative for chronic obstructive lung disease airflow stage of 3 (severe) [2]. See Table 1 for additional participant characteristics. The groups differed in baseline characteristics in terms of use of gait aid ($p < 0.001$) with more study group participants using one, self-reported general health ($p < 0.001$) with study group participants reporting worse health, and comorbidities with more study group participants having anxiety ($p < 0.001$) and depression ($p = 0.047$) and more control group participants having cataracts ($p = 0.046$).

Other than the social subscale, the LLDI scores were not normally distributed for at least one of the groups. Therefore, non-parametric tests were conducted. We calculated the probability of an observation in the control group having a true value higher than an observation in the study group [18]. Both LLDI domain scores and all four subscale scores were significantly higher for the control group than the study group with probabilities ranging from 0.63 to 0.74 (Table 2).

These results were similar when the LLDI scores of the full sample of 96 people with COPD and 202 people without respiratory disease were compared (see S1 and S2 Tables in S2 File).

**Table 2. Between-group comparison of LLDI scores.**

| | Study group, median (25th-75th percentile) | Control group, median (25th-75th percentile) | Between-group comparison, z (p) | Probability (95% CI) that control group score > study group score |
|---|---|---|---|---|
| **Frequency domain** | 47.2 (44.5–51.5) | 52.6 (46.9–58.0) | -4.18 (<0.001) | 0.72 (0.63–0.80) |
| **Personal subscale** | 51.7 (47.9–62.8) | 65.1 (56.3–84.0) | -4.53 (<0.001) | 0.74 (0.64–0.83) |
| **Social subscale** | 41.1 (37.3–46.3) | 46.3 (40.5–53.3) | -3.13 (0.011) | 0.66 (0.57–0.76) |
| **Limitation domain** | 59.7 (51.8–64.8) | 67.3 (58.1–83.4) | -4.34 (<0.001) | 0.73 (0.64–0.81) |
| **Instrumental subscale** | 55.4 (48.5–64.4) | 65.5 (55.8–88.9) | -4.46 (<0.001) | 0.73 (0.65–0.82) |
| **Management subscale** | 78.3 (71.0–100.0) | 89.5 (74.5–100.0) | -2.64 (0.05) | 0.63 (0.54–0.73) |

LLDI = late life disability instrument.

The difference in median scores for the frequency and limitation domains for the two groups was lower than the $MDC_{95}$ established in people with COPD (5.4 points compared to 6.7 points, and 7.6 points compared to 9.9 points, respectively) and the $MDC_{90}$ established in mobility-limited older adults for the frequency and limitation domains (7.4 and 11.6 points, respectively) [13,16]. The largest magnitude of difference between groups was seen for the personal subscale of the frequency domain (median difference 13.4 points), followed by the management and instrumental subscales of the limitation domain (11.2 and 10.1 points, respectively), with the smallest magnitude of difference being for the social subscale of the frequency domain (median difference 5.2 points). Figs 2 and 3 show the distribution of the frequency and limitation domain scores for both groups.

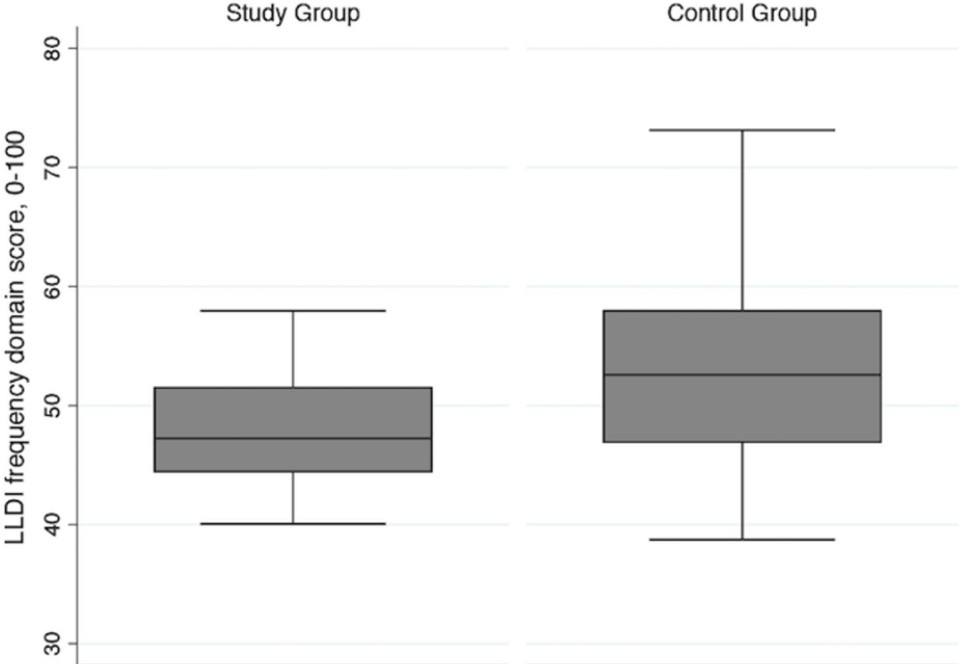

**Fig 2. Late-life disability instrument frequency domain scores for the study group and control group.** The box represents the 25th to 75th percentiles with the horizontal line inside the box representing the median score. The horizontal lines above and below the box represent the maximum (75th percentile plus 1.5*IQR) and minimum (25th percentile minus 1.5*IQR) scores. IQR = interquartile range.

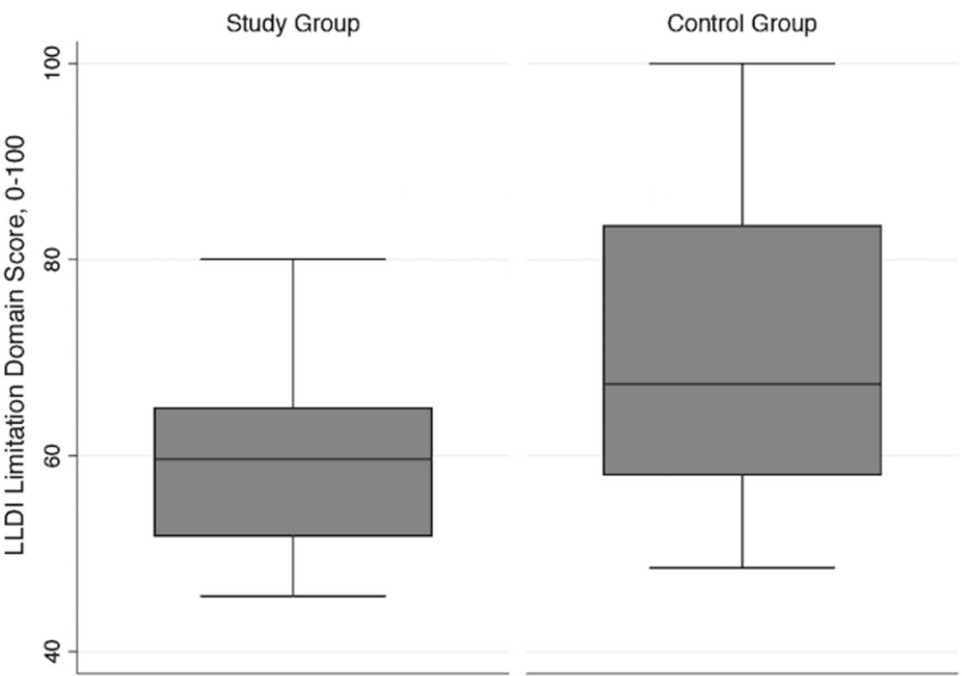

**Fig 3. Late-life disability instrument limitation domain scores for the study group and control group.** The box represents the 25th to 75th percentiles with the horizontal line inside the box representing the median score. The horizontal lines above and below the box represent the maximum (75th percentile plus 1.5*IQR) and minimum (25th percentile minus 1.5*IQR) scores. IQR = interquartile range.

## Discussion

This is the first study to compare scores on a validated measure of participation between people with COPD and an age- and sex-matched sample of older adults without respiratory disease. The results showed, as hypothesized, that people with COPD have greater participation restrictions than age-matched adults. They participated in tasks less frequently and had greater limitations in their ability to participate in life situations, particularly in those that involved some level of mobility or physical function such as taking care of the home and active recreation.

The greatest magnitude of difference between the median scores for the study group and the control group were seen for the personal role subscale of the frequency domain and the management role subscale of the limitation domain. Life tasks represented in both of these subscales are taking care of household business and finances and taking care of one's own health (i.e., how often do you take care of your own health and how limited do you feel in taking care of your own health). There was also a large magnitude of difference in limitations related to the instrumental role which includes tasks such as taking part in a regular fitness program, taking care of one's own personal care needs, taking care of local errands, and preparing meals for oneself, all tasks that require some level of mobility or physical activity [12] which is a recognized limitation in people with COPD [2].

A smaller difference, and potentially a less clinically meaningful difference, in scores between groups was seen for the social subscale of the frequency domain, suggesting that the two groups experienced similar reductions in social activities. This subscale, comprised of tasks such as visiting friends and family, volunteering, travelling, going out with others to public places, and participating in organized social activities, was the lowest scoring subscale

(below 50%) for both groups. The control group completed the questionnaire at the end of the first wave of the COVID-19 pandemic when social and public health restrictions were in place whereas all of the people from the study group participated prior to the pandemic. Accordingly, the control groups' frequency of participating in these life tasks was likely impacted, highlighting the extent to which participation is restricted for older adults during a pandemic and potentially explaining the smaller difference between groups in this subscale. Normative scores have not been established for the LLDI but two previous studies of community-dwelling adults aged 65 and over reported mean scores of 41.4 [19] and 45.5 [20] points on the social subscale, similar to the mean score of 47.2 points seen in our control group. While the magnitude of difference on the social subscale between the study group and control group was less than that of the other subscales, these tasks should remain a focus of intervention for all older adults, given that both groups scored < 50%.

Some additional variability and between-group differences may have been missed as the limitation domain showed a ceiling effect [21] for the control group with 21% scoring 100%. In the initial study of the development of the measure, only 6.7% of the respondents (adults ≥ 60 years with a range of functional limitations) scored 100% on the limitation domain [22]. Subsequent studies in general populations of older adults have varied from no ceiling effect for the limitation domain [23,24], to > 30% of participants scoring 100% [25]. These inconsistent findings are likely due to differences in age (mean age ≥ 79 [23,24] vs 69 years [25]) and physical activity levels of the participants, but they are worthy of further exploration. In particular, we noted a ceiling effect on the management subscale of the limitation domain, with 26% of the study group participants and 43% of the control group participants scoring 100%. It is therefore likely that this measure has not captured the full range of limitations related to communication and planning (or non-mobility-related life tasks) in people with COPD.

The wide distribution of scores, particularly in the control group, is important to note. The control group was a random sample of older adults who were only excluded from the tele-survey if they were unable to complete the questionnaires. Therefore, as is expected in the general population of older adults [26], the majority of participants had multimorbidity. For the purpose of this analysis, we excluded those with respiratory diagnoses, but there were people with arthritis, diabetes and vision impairment, all diagnoses that have a potential impact on participation [9]. We likely would have seen a greater magnitude of difference in median scores had the control group been a healthy group without any chronic conditions. In addition, the spread of the scores in the control group likely reflected the within-group heterogeneity associated with differences in intrinsic factors (e.g., health conditions and functional impairments) and extrinsic factors (e.g., medications, clinical treatments, assistive devices, and barriers in the built environment) [27] less likely in the more homogenous group with COPD.

The decreased frequency of and increased limitations in participation found in people with COPD highlights the importance of assessing this important aspect of health in this population and addressing participation restrictions as part of pulmonary rehabilitation in order to mitigate negative sequalae such as psychological distress and mortality.

## Limitations

The LLDI scores demonstrated by the study group reflect those with severe COPD and therefore cannot be generalized to those with mild disease. This was a secondary analysis with data retrieved from two independent studies in which the LLDI was measured. As the primary studies differed in design, we did not consistently have other baseline data or outcome measures, such as physical function or quality of life, that might have more completely

characterized the participants. In addition, we did not have pulmonary function test results from the control group participants and it is possible that some of them had undiagnosed respiratory disease. And finally, the control group completed the study at the end of the first wave of the COVID-19 pandemic at which time there were social and public health restrictions in place. This difference in social circumstances might have impacted the frequency of participation in life tasks, making the control group results non-generalizable to non-pandemic times. The frequency of social activities was low in both groups and it is likely that the control group would have shown even higher scores on this subscale had public health restrictions not been in place at the time. As such, the differences seen in the participation scores between the two groups may have been attenuated due to temporal confounding.

## Conclusion

People with COPD have greater restrictions in both their frequency of participation and their limitations in participation, than their peers without respiratory disease, especially in life tasks related to personal, management, and instrumental roles. Clinicians have a unique opportunity to address the extra-pulmonary effects of COPD in pulmonary rehabilitation programs. Valid measures of participation are not generally included in such programs that address the longer-term impact of COPD on patients and their families. However, an assessment of participation restrictions could be a valuable addition to the management of chronic lung disease so that targeted interventions can be considered for these patients.

## Supporting information

**S1 File. Raw data for study group and control group.** Group 1 = study group; Group 2 = control group.
(XLSX)

**S2 File. Participant characteristics and between-group comparison of LLDI scores for 96 study group participants and 202 control group participants who were not age- or sex-matched.**
(DOCX)

## Author Contributions

**Conceptualization:** Sachi O'Hoski, Ayse Kuspinar, Joshua Wald, Julie Richardson, Roger Goldstein, Marla K. Beauchamp.

**Formal analysis:** Sachi O'Hoski.

**Investigation:** Sachi O'Hoski.

**Methodology:** Sachi O'Hoski, Ayse Kuspinar, Joshua Wald, Julie Richardson, Roger Goldstein, Marla K. Beauchamp.

**Project administration:** Sachi O'Hoski.

**Resources:** Ayse Kuspinar, Joshua Wald, Julie Richardson, Roger Goldstein, Marla K. Beauchamp.

**Supervision:** Ayse Kuspinar, Joshua Wald, Julie Richardson, Roger Goldstein, Marla K. Beauchamp.

**Writing – original draft:** Sachi O'Hoski.

**Writing – review & editing:** Sachi O'Hoski, Ayse Kuspinar, Joshua Wald, Julie Richardson, Roger Goldstein, Marla K. Beauchamp.

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
