## [Decision Letter · Decision Letter 0]

14 Jun 2022

PONE-D-22-03462People with chronic obstructive pulmonary disease have greater participation restrictions than their peersPLOS ONE

Dear Dr. Maria K Beauchamp, 

Thank you for submitting your manuscript to PLOS ONE. After careful consideration, we feel that it has merit but does not fully meet PLOS ONE’s publication criteria as it currently stands. Therefore, we invite you to submit a revised version of the manuscript that addresses the points raised during the review process.

 Response from editor:Please kindly review the comment by different reviewers as below, which I believe will improve the quality of the manuscript. I also have few major concerns about this study:

1. The title about comparing the limitations between elderly with COPD and their peers is inaccurate. The control group is the elderly assessed during the COVID-19 pandemic. Covid-19 pandemic per se is a cofounding factor. Please make corrections as per suggestion.

2. The baseline demographic even match, is not identical, and not every characteristic matches. Therefore, the comparison must be made and corrections/adjustments in the statistical analysis of study outcomes should be made if any significant difference is noted.

3. The sample size is small, therefore, abnormal distribution is expected. Authors also declare the distribution of primary outcomes is abnormal. Thus, the continuous variables have to be reported in the median, even the result of reporting the mean is also no different. Same as the statistic methods.

4. The fact that the study was conducted at different time frames is a major limitation. Please justify that.

5. Do your sample size adequate?

Hope all these points will help with your correction.

We look forward to receiving your revised manuscript.

Kind regards,

Chee-Shee Chai, MD, MMed, FCCP,

Academic Editor

PLOS ONE

Journal Requirements:

Reviewers' comments:

Reviewer's Responses to Questions

**Comments to the Author**

1. Is the manuscript technically sound, and do the data support the conclusions?

Reviewer #1: Yes

Reviewer #2: Yes

Reviewer #3: Yes

2. Has the statistical analysis been performed appropriately and rigorously? 

Reviewer #1: Yes

Reviewer #2: Yes

Reviewer #3: Yes

3. Have the authors made all data underlying the findings in their manuscript fully available?

Reviewer #1: Yes

Reviewer #2: Yes

Reviewer #3: Yes

4. Is the manuscript presented in an intelligible fashion and written in standard English?

Reviewer #1: Yes

Reviewer #2: Yes

Reviewer #3: Yes

5. Review Comments to the Author

Reviewer #1: This is an interesting study that compares limitations to participation in a cohort of subjects with advanced COPD against age and sex matched control subjects. I would recommend the following revisions be undertaken:

Major comments:

1) One of the major limitations of this study is that it is temporally confounded. The control group completed the questionnaire during the first wave of the COVID-19 pandemic when social and public health restrictions were in place, whereas all of the people from the study group participated prior to the pandemic. Accordingly, the control groups’ frequency of participating in these life tasks was likely impacted, highlighting the extent to

which participation is restricted for older adults during a pandemic. The overall effect of this confounding would have been to reduce inter-group differences seen on the LLDI scores. I think that this major limitation has to be mentioned and explained within the abstract of the paper, and the abstract has to conclude that 'People with COPD had greater participation restrictions than a random sample of older adults without ongoing respiratory conditions, but differences in participation restriction seen in this study may have been reduced due to temporal confounding.

2) The authors state on page 6 that "The minimal detectable change score (MDC90) for the

LLDI is 7.4 points for the frequency scale and 11.6 points for the limitation scale in mobility limited older adults." Since the between-group differences in this study were less than these MDC90 scores does this mean that the differences are not clinically significant between groups? Please clarify.

3) Did the control group undergo lung function testing? If not, how can we be sure they did not have COPD?

4) A Figure 1 flow diagram needs to be added which shows how the study subjects were recruited and included/excluded and which subjects were ultimately included in the final study analysis.

5) I note that from the primary studies, LLDI scores were collected in 96 people with COPD and 272

older adults. For this paper, only 46 study group participants and 92 control group participants were retained. This means that a lot of interesting collected data has not been analyzed. I think it would be interesting to see the LLDI scores for the entire COPD cohort and the entire 272 control cohort. This analysis could be added to the paper, and the between-group differences could be adjusted for age and sex. This additional age and sex adjusted analysis would be complementary to the case-control analysis.

6) Minor comment- please delete the dots showing outliers from figs 1 and 2- they are distracting and not needed.

Reviewer #2: This is a well-written manuscript, an interesting and relevant work on participation in people with chronic obstructive pulmonary disease (COPD). The authors have compared the level of participation between people with COPD and community-dwelling older adults without lung disease. People with COPD presented with greater participation restrictions compared to their peers without chronic respiratory conditions. Please find below some major and minor concerns the authors should address before publication.

Abstract

Minor

The LLDI abbreviation should be indicated at its first appearance in the text.

Results

Minor

In Table 1, the presentation of statistical differences between groups should be revised to support the manuscript text. Some symbols used should be moved to the first column for a description of the variables, and only statistical differences should be indicated in the middle and left data columns. It is not clear to the reader which frequencies of self-reported general health differences are between groups.

Major

In the study group, is there any subanalysis available on differences in LLDI Participation scores between people with COPD on Long-term home oxygen therapy (LTOT) compared to those who are not? This would bring new information to the COPD literature. The authors should also provide in Table 1 the number of study group participants on LTOT, as it could impact Participation due to the use of non-portable oxygen delivery devices.

Discussion

Page 13, line 225: please consider the replacement of 'pathologies' for 'diseases' or 'illnesses'.

Page 14, lines 244-247: the authors should consider rewriting the sentences as it is difficult for the reader knows who they are referring to in 'We have a unique opportunity...' Are the authors referring to themselves, physiotherapists or healthcare professionals in general involved in COPD rehabilitation? Also, the study sample comprised patients who were not exclusively on admission to pulmonary rehabilitation programs.

Reviewer #3: Sachi O’Hoski et al s aimed to compare participation scores in people with COPD to scores from a random sample of older adults using a validated measure of participation, the LLDI which is consistent with the concept of participation in the international classification of functioning, disability and health. LLDI sores were lower for the COPD group compared to controls for both the frequency and limitation domains of the LLDI. The paper is generally written in an excellent way, the description of methods used is clear, results are nicely presented and discussion is focused. However, the study is limited by its design.

I have the following concerns.

1. The authors used a secondary analysis of existing data to compare COPD patients and controls in terms of quality-of-life outcomes. Data are derived from two different time periods, one of the them during the COVID-19 pandemic. This type of design may have introduced biases in the accuracy of the results of this study.

2. During the pandemic many things occurred in terms of our living that may have affected subjects’ decisions and activities. In this respect activities and participation may have been different compared to the previous period- where COPD patients were recruited. How the authors treated this source of potential bias in their analysis? Has LLDI been validated during these circumstances? Or, do we know that this instrument is accurate even during periods with such different social conditions? It would be useful to know that LLDI provides similar results in similar populations but in different time periods/social conditions (or not).

3. In addition, COPD patients experience relatively severe disease based on the FEV1 reported. Is there a similar assessment of the respiratory function for controls?

4. Introduction could be shorter. I would highlight more -being more specific- the clinical value of the results of this study in the discussion section

---

## [Author Response · Author response to Decision Letter 0]

12 Jul 2022

Editor:

1. The title about comparing the limitations between elderly with COPD and their peers is inaccurate. The control group is the elderly assessed during the COVID-19 pandemic. Covid-19 pandemic per se is a cofounding factor. Please make corrections as per suggestion.

Response (R): The title has been changed to “People with COPD have greater participation restrictions than age-matched older adults without respiratory conditions assessed during the COVID-19 pandemic” and we have removed ‘peers’ throughout the manuscript and instead referred to the control group as “older adults of the same age without respiratory conditions” (Page 2 Line 26) or “age-matched adults” (Page 13 Line 204). We have also acknowledged that the age-matched group was assessed during the COVID-19 pandemic in the abstract on Page 2 Lines 31-33: “The study group completed the LLDI prior to the COVID-19 pandemic and the control group completed the LLDI during the first wave of the pandemic.” and Pages 2-3 Lines 44-45: “The differences seen in participation between the two groups may have been reduced due to temporal confounding from the COVID-19 pandemic.” We have also expanded on this limitation in the discussion on Pages 15-16 Lines 268-276: “And finally, the control group completed the study during the first wave of the COVID-19 pandemic at which time there were social and public health restrictions in place. This difference in social circumstances might have impacted the frequency of participation in life tasks, making the control group results non-generalizable to non-pandemic times. The frequency of social activities was low in both groups and it is likely that the control group would have shown even higher scores on this subscale had public health restrictions not been in place at the time. As such, the differences seen in the participation scores between the two groups may have been attenuated due to temporal confounding.”

2. The baseline demographic even match, is not identical, and not every characteristic matches. Therefore, the comparison must be made and corrections/adjustments in the statistical analysis of study outcomes should be made if any significant difference is noted.

R: We included a narrative summary of the differences between groups on Page 8 Lines 159-163: “The groups differed in baseline characteristics in terms of use of gait aid (p < 0.001) with more study group participants using one, self-reported general health (p < 0.001) with study group participants reporting worse health, and comorbidities with more study group participants having anxiety (p < 0.001) and depression (p = 0.047) and more control group participants having cataracts (p = 0.046).” We have now added a column to Table 1 with the statistical comparison of baselines characteristics between groups. We did not adjust for these differences between groups in our analyses because these differences in the characteristics of people with COPD are likely related to their increased participation restrictions. We were not solely interested in the impact of a diagnosis of COPD on participation, but in the impact of having COPD and all that goes along with it, including use of a gait aid, lower self-reported health, and comorbidities such as anxiety and depression. 

3. The sample size is small, therefore, abnormal distribution is expected. Authors also declare the distribution of primary outcomes is abnormal. Thus, the continuous variables have to be reported in the median, even the result of reporting the mean is also no different. Same as the statistic methods.

R: We have adjusted the reporting of all variables and outcomes to account for the non-normal distribution of the LLDI scores. Changes to the manuscript can be found in the following sections:

Abstract methods and results on Page 2 Lines 34 and 37-42, respectively.

Page 8 Line 157: Results reported as median (25th-75th percentile)

Page 9 (Table 1): Results reported as median (25th-75th percentile)

Pages 10-11 Lines 170-174: “Other than the social subscale, the LLDI scores were not normally distributed for at least one of the groups. Therefore, non-parametric tests were conducted. We calculated the probability of an observation in the control group having a true value higher than an observation in the study group [18]. Both LLDI domain scores and all four subscale scores were significantly higher for the control group than the study group with probabilities ranging from 0.63 to 0.74 (Table 2).” 

Page 11 (Table 2): Median (25th-75th percentile) and probabilities added to the table.

Page 12 Lines 184-188: “The largest magnitude of difference between groups was seen for the personal subscale of the frequency domain (median difference 13.4 points), followed by the management and instrumental subscales of the limitation domain (11.2 and 10.1 points, respectively), with the smallest magnitude of difference being for the social subscale of the frequency domain (median difference 5.2 points).”

Page 13 Lines 207-212: We have adjusted this part of the discussion since the subscales with the largest mean difference and the largest median difference were different. 

4. The fact that the study was conducted at different time frames is a major limitation. Please justify that.

R: Given that this was a secondary analysis, we were unable to control when the data were collected. However, this has been included as a limitation on Pages 15-16 Lines 268-276: “And finally, the control group completed the study during the first wave of the COVID-19 pandemic at which time there were social and public health restrictions in place. This difference in social circumstances might have impacted the frequency of participation in life tasks, making the control group results non-generalizable to non-pandemic times. The frequency of social activities was low in both groups and it is likely that the control group would have shown even higher scores on this subscale had public health restrictions not been in place at the time. As such, the differences seen in the participation scores between the two groups may have been attenuated due to temporal confounding.”

5. Do your sample size adequate?

R: Because this was a secondary analysis, we did not calculate a sample size a priori. For the frequency domain, assuming a pooled standard deviation of 10, power of 80% and alpha of 5%, we would need 38 participants in the study group and 76 participants in the control group to detect a difference of 5 points. For the limitation domain, assuming a pooled standard deviation of 15, power of 80% and alpha of 5%, we would need 44 participants in the study group and 88 participants in the control group to detect a difference of 9 points. With 46 study group participants and 92 control group participants, we were powered to detect clinically important differences. 

R: We have made adjustments to the formatting to ensure the manuscript meets the style requirements. 

7. We note that the grant information you provided in the ‘Funding Information’ and ‘Financial Disclosure’ sections do not match. 

R: We have corrected the financial disclosure section so that it matches the funding information. 

We have added captions for the S1 Full Sample Analysis and S2 Data File at the end of the manuscript and referred to them in the text where appropriate.

Reviewer #1: 

This is an interesting study that compares limitations to participation in a cohort of subjects with advanced COPD against age and sex matched control subjects. I would recommend the following revisions be undertaken:

Major comments:

1) One of the major limitations of this study is that it is temporally confounded. The control group completed the questionnaire during the first wave of the COVID-19 pandemic when social and public health restrictions were in place, whereas all of the people from the study group participated prior to the pandemic. Accordingly, the control groups’ frequency of participating in these life tasks was likely impacted, highlighting the extent to

which participation is restricted for older adults during a pandemic. The overall effect of this confounding would have been to reduce inter-group differences seen on the LLDI scores. I think that this major limitation has to be mentioned and explained within the abstract of the paper, and the abstract has to conclude that 'People with COPD had greater participation restrictions than a random sample of older adults without ongoing respiratory conditions, but differences in participation restriction seen in this study may have been reduced due to temporal confounding.

R: As suggested, we have added to the methods section of the abstract on Page 2 Lines 31-33: “The study group completed the LLDI prior to the COVID-19 pandemic and the control group completed the LLDI during the first wave of the pandemic.” as well as to the abstract conclusion on Pages 2-3 Lines 44-45: “The differences seen in participation between the two groups may have been reduced due to temporal confounding from the COVID-19 pandemic.” 

2) The authors state on page 6 that "The minimal detectable change score (MDC90) for the

LLDI is 7.4 points for the frequency scale and 11.6 points for the limitation scale in mobility limited older adults." Since the between-group differences in this study were less than these MDC90 scores does this mean that the differences are not clinically significant between groups? Please clarify.

R: The reviewer is correct that the MDC90 values reported in a study of mobility limited older adults were 7.4 and 11.6 points for frequency and limitation, respectively. However, it is also important to note that MDC values are population-specific. We have previously established MDC95 values in people with COPD that were lower than what was reported in the study of older adults (4.82 and 7.21 points for frequency and limitation, respectively) (O’Hoski S, et al. A tool to assess participation in people with COPD. Chest. 2021;159(1):138-146.). We have added reference to these values into the manuscript on Page 7 Lines 132-133: “…and the MDC95 is 4.8 points for the frequency scale and 7.2 points for the limitation scale in people with COPD [13].” We have updated the results on Pages 12 Lines 180-184 to reflect both sets of previously established MDC values: “The difference in median scores for the frequency and limitation domains for the two groups was higher than the MDC95 established in people with COPD (5.4 points compared to 4.8 points, and 7.6 points compared to 7.2 points, respectively) but lower than the MDC90 established in mobility-limited older adults for the frequency and limitation domains (7.4 and 11.6 points, respectively) [13,16].” The magnitude of differences seen in participation between the groups may have been reduced due to temporal confounding as previously mentioned. 

3) Did the control group undergo lung function testing? If not, how can we be sure they did not have COPD?

R: Unfortunately, we do not have pulmonary function test data for the control group participants. We have updated the manuscript to indicate that control group participants were those who did not “report having respiratory conditions” (Page 2 Lines 30-31) and that we excluded participants if they had “physician-diagnosed respiratory condition” (Page 8 Line 151). We have also added this as a limitation on Pages 15-16 Lines 267-268: “In addition, we did not have pulmonary function test results from the control group participants and it is possible that some of them had undiagnosed respiratory disease.”

4) A Figure 1 flow diagram needs to be added which shows how the study subjects were recruited and included/excluded and which subjects were ultimately included in the final study analysis.

R: A flow diagram (now Figure 1) has been added to show recruitment, inclusion and exclusion. 

5) I note that from the primary studies, LLDI scores were collected in 96 people with COPD and 272 older adults. For this paper, only 46 study group participants and 92 control group participants were retained. This means that a lot of interesting collected data has not been analyzed. I think it would be interesting to see the LLDI scores for the entire COPD cohort and the entire 272 control cohort. This analysis could be added to the paper, and the between-group differences could be adjusted for age and sex. This additional age and sex adjusted analysis would be complementary to the case-control analysis.

R: Of the 272 participants in the general sample of older adults, there were 202 who had complete scores for the LLDI and did not have a respiratory condition. The full sample (96 study group participants and 202 control group participants) was compared using quantile regression including age and sex as covariates and having COPD was a significant predictor of both frequency of and limitations in participation. We have added to the methods on Page 8 Lines 144-146: "As a sensitivity analysis, we also compared LLDI scores between the full samples of people with and without respiratory disease, without age- and sex-matching using quantile regression with age and sex as covariates." We have also added to the results on Page 12 Lines 177-179: "These results were similar when the LLDI scores of the full sample of 96 people with COPD and 202 people without respiratory disease were compared (see Supplementary Tables 1 and 2 in S2 Full sample analysis)."

6) Minor comment- please delete the dots showing outliers from figs 1 and 2- they are distracting and not needed.

R: Figures 2 and 3 (previously figures 1 and 2) have been revised as requested. 

Reviewer #2: 

This is a well-written manuscript, an interesting and relevant work on participation in people with chronic obstructive pulmonary disease (COPD). The authors have compared the level of participation between people with COPD and community-dwelling older adults without lung disease. People with COPD presented with greater participation restrictions compared to their peers without chronic respiratory conditions. Please find below some major and minor concerns the authors should address before publication.

Abstract, Minor

1. The LLDI abbreviation should be indicated at its first appearance in the text.

R: This has been added (Page 2 Line 27). 

Results, Minor

2. In Table 1, the presentation of statistical differences between groups should be revised to support the manuscript text. Some symbols used should be moved to the first column for a description of the variables, and only statistical differences should be indicated in the middle and left data columns. It is not clear to the reader which frequencies of self-reported general health differences are between groups.

R: Table 1 on Pages 9-10 has been revised to include statistical comparisons between the two groups so that column 2 presents the study group data, column 3 present the control group data and column 4 presents the between-group p-values. 

Results, Major

3. In the study group, is there any subanalysis available on differences in LLDI Participation scores between people with COPD on Long-term home oxygen therapy (LTOT) compared to those who are not? This would bring new information to the COPD literature. The authors should also provide in Table 1 the number of study group participants on LTOT, as it could impact Participation due to the use of non-portable oxygen delivery devices.

R: We have added supplemental oxygen use for the study group to Table 1 on Page 9. While the use of supplemental oxygen may impact participation frequency and limitations, we chose not to explore it in the current paper as it has been previously investigated in another secondary analysis involving this data (D’Amore C, O’Hoski S, Griffith LE, Richardson J, Goldstein RS, Beauchamp MK. Factors associated with participation in life situations in people with COPD. Chron Respir Dis. 2022;19:1–8. DOI: 10.1177/14799731221079305.)

Discussion, Minor

4. Page 13, line 225: please consider the replacement of 'pathologies' for 'diseases' or 'illnesses'.

R: This has been changed to “health conditions” (Page 15 Line 254).

5. Page 14, lines 244-247: the authors should consider rewriting the sentences as it is difficult for the reader knows who they are referring to in 'We have a unique opportunity...' Are the authors referring to themselves, physiotherapists or healthcare professionals in general involved in COPD rehabilitation? Also, the study sample comprised patients who were not exclusively on admission to pulmonary rehabilitation programs.

R: We have changed this sentence on Page 16 Lines 280-282 to, “Clinicians have a unique opportunity…” The reviewer is correct that our sample was not exclusively PR participants. Our intended meaning here is that measurement of participation may be a valuable addition to these programs because of the lower scores seen among people with COPD. There is a need to assess participation in people with COPD and to treat restrictions and PR programs might be the place to do that since patients are followed for an extended period of time. Clinicians in PR have a unique opportunity to address the longer-term consequences of the disease. 

Reviewer #3: 

Sachi O’Hoski et al aimed to compare participation scores in people with COPD to scores from a random sample of older adults using a validated measure of participation, the LLDI which is consistent with the concept of participation in the international classification of functioning, disability and health. LLDI sores were lower for the COPD group compared to controls for both the frequency and limitation domains of the LLDI. The paper is generally written in an excellent way, the description of methods used is clear, results are nicely presented and discussion is focused. However, the study is limited by its design.

I have the following concerns.

1. The authors used a secondary analysis of existing data to compare COPD patients and controls in terms of quality-of-life outcomes. Data are derived from two different time periods, one of the them during the COVID-19 pandemic. This type of design may have introduced biases in the accuracy of the results of this study.

R: We have added that the control group’s social participation was likely impacted by the pandemic, “potentially explaining the smaller difference between groups in this subscale” (Page 14 Lines 226-227). We have also indicated this as a limitation on Page 16 Lines 268-276: “And finally, the control group completed the study during the first wave of the COVID-19 pandemic at which time there were social and public health restrictions in place. This difference in social circumstances might have impacted the frequency of participation in life tasks, making the control group results non-generalizable to non-pandemic times. The frequency of social activities was low in both groups and it is likely that the control group would have shown even higher scores on this subscale had public health restrictions not been in place at the time. As such, the differences seen in the participation scores between the two groups may have been attenuated due to temporal confounding.”

2. During the pandemic many things occurred in terms of our living that may have affected subjects’ decisions and activities. In this respect activities and participation may have been different compared to the previous period- where COPD patients were recruited. How the authors treated this source of potential bias in their analysis? Has LLDI been validated during these circumstances? Or, do we know that this instrument is accurate even during periods with such different social conditions? It would be useful to know that LLDI provides similar results in similar populations but in different time periods/social conditions (or not).

R: We have acknowledged that participation scores for the control group may have been higher had it not been for the pandemic and public health restrictions and indicated that this was a potential limitation. We have not adjusted our analyses for this temporal difference as our aim was to simply investigate the difference in scores between the groups. The LLDI has strong evidence for its psychometric properties in various populations and contexts (Beauchamp MK, Schmidt CT, Pedersen MM, Bean JF, Jette AM. Psychometric properties of the Late-Life Function and Disability Instrument: A systematic review. BMC Geriatr 2014;14:12. doi: 10.1186/1471-2318-14-12) and while scores may differ based on a variety of factors such as external barriers to participation and participant choice, these factors are individual and are taken into account when one is answering the questions on the LLDI. While we have assumed that scores on some of the subscales would have been impacted by social restrictions, this should not impact the validity of the measure. In terms of comparison of scores in different social conditions, we have provided some information about scores found in the literature pre-pandemic on Pages 14 Lines 227-230: “Normative scores have not been established for the LLDI but two previous studies of community-dwelling adults aged 65 and over reported mean scores of 41.4 [19] and 45.5 [20] points on the social subscale, similar to the mean score of 47.2 points seen in our control group.” 

3. In addition, COPD patients experience relatively severe disease based on the FEV1 reported. Is there a similar assessment of the respiratory function for controls?

R: We do not have data on respiratory function for the control group. This was a secondary analysis of data so we were limited by what information was collected for the primary study. We have added this as a limitation on Pages 15-16 Lines 267-268: “In addition, we did not have pulmonary function test results from the control group participants and it is possible that some of them had undiagnosed respiratory disease.”

4. Introduction could be shorter. I would highlight more -being more specific- the clinical value of the results of this study in the discussion section

R: We have added to the discussion on Page 15 Lines 257-260: “The decreased frequency of and increased limitations in participation found in people with COPD highlights the importance of assessing this important aspect of health in this population and addressing participation restrictions as part of pulmonary rehabilitation in order mitigate negative sequalae such as psychological distress and mortality.”

---

## [Decision Letter · Decision Letter 1]

13 Sep 2022

People with COPD have greater participation restrictions than age-matched older adults without respiratory conditions assessed during the COVID-19 pandemic

PONE-D-22-03462R1

Dear Dr. Beauchamp,

We’re pleased to inform you that your manuscript has been judged scientifically suitable for publication and will be formally accepted for publication once it meets all outstanding technical requirements.

Kind regards,

Wanich Suksatan

Academic Editor

PLOS ONE

Additional Editor Comments (optional):

Congratulations for successful amendments!

---

## [Editor Report · Acceptance letter]

26 Sep 2022

PONE-D-22-03462R1 

People with COPD have greater participation restrictions than age-matched older adults without respiratory conditions assessed during the COVID-19 pandemic 

Dear Dr. Beauchamp:

I'm pleased to inform you that your manuscript has been deemed suitable for publication in PLOS ONE. Congratulations! Your manuscript is now with our production department. 

Kind regards, 

on behalf of

Dr. Wanich Suksatan 

Academic Editor

PLOS ONE